# Baseline Gut Metagenomic Functional Gene Signature Associated with Variable Weight Loss Responses following a Healthy Lifestyle Intervention in Humans

Christian Diener,[a] Shizhen Qin,[a] Yong Zhou,[a] Sushmita Patwardhan,[a] Li Tang,[a] Jennifer C. Lovejoy,[a,b] Andrew T. Magis,[a] Nathan D. Price,[a,c,d] Leroy Hood,[a,c] Sean M. Gibbons[a,c,e]

[a]Institute for Systems Biology, Seattle, Washington, USA
[b]Lifestyle Medicine Institute, Redlands, California, USA
[c]Department of Bioengineering, University of Washington, Seattle, Washington, USA
[d]Onegevity (a division of Thorne HealthTech), New York, New York, USA
[e]eScience Institute, University of Washington, Seattle, Washington, USA

**ABSTRACT** Recent human feeding studies have shown how the baseline taxonomic composition of the gut microbiome can determine responses to weight loss interventions. However, the functional determinants underlying this phenomenon remain unclear. We report a weight loss response analysis on a cohort of 105 individuals selected from a larger population enrolled in a commercial wellness program, which included healthy lifestyle coaching. Each individual in the cohort had baseline blood metabolomics, blood proteomics, clinical labs, dietary questionnaires, stool 16S rRNA gene sequencing data, and follow-up data on weight change. We generated additional targeted proteomics data on obesity-associated proteins in blood before and after intervention, along with baseline stool metagenomic data, for a subset of 25 individuals who showed the most extreme weight change phenotypes. We built regression models to identify baseline blood, stool, and dietary features associated with weight loss, independent of age, sex, and baseline body mass index (BMI). Many features were independently associated with baseline BMI, but few were independently associated with weight loss. Baseline diet was not associated with weight loss, and only one blood analyte was associated with changes in weight. However, 31 baseline stool metagenomic functional features, including complex polysaccharide and protein degradation genes, stress-response genes, respiration-related genes, and cell wall synthesis genes, along with gut bacterial replication rates, were associated with weight loss responses after controlling for age, sex, and baseline BMI. Together, these results provide a set of compelling hypotheses for how commensal gut microbiota influence weight loss outcomes in humans.

**IMPORTANCE** Recent human feeding studies have shown how the baseline taxonomic composition of the gut microbiome can determine responses to dietary interventions, but the exact functional determinants underlying this phenomenon remain unclear. In this study, we set out to better understand interactions between baseline BMI, metabolic health, diet, gut microbiome functional profiles, and subsequent weight changes in a human cohort that underwent a healthy lifestyle intervention. Overall, our results suggest that the microbiota may influence host weight loss responses through variable bacterial growth rates, dietary energy harvest efficiency, and immunomodulation.

**KEYWORDS** amylase, diet, health, metabolome, metagenome, microbiome, obesity, proteome, replication rate, weight loss

Address correspondence to Christian Diener, christian.diener@isbscience.org, or Sean M. Gibbons, sgibbons@isbscience.org.

🐦 Baseline gut microbiome functional signature associated with weight loss response following a healthy lifestyle intervention, independent of initial BMI or metabolic health state.

The question of whether or not the ecological composition of the gut microbiome may causally affect weight loss remains somewhat controversial (1). The composition and diversity of the gut microbiome have been correlated with body mass index (BMI) and metabolic health markers and have been shown to modulate weight gain in mice (2). There are many confounding variables with regard to obesity phenotypes, including genetics, prior health status, age, physical activity, and diet, which can modulate whether or not a person who is nominally "overweight" or "obese" is considered metabolically healthy (3–6). Recent work by our group has demonstrated that certain host-microbe metabolic associations are disrupted only in individuals experiencing severe obesity (BMI $\geq$ 35), but not in overweight or mildly obese individuals (25 < BMI < 35), relative to individuals with a normal weight (BMI < 25) (7). Thus, the cutoffs used to define obesity may not always match the underlying metabolism- and microbiome-associated phenotypic heterogeneity in the population.

Whether or not a consistent association between the microbiota and obesity phenotypes exists, another important, but unresolved, question is whether or not the human gut microbiome contributes directly to changes in weight after an intervention, independent of baseline BMI. While the gut microbiome has been shown to contribute to weight gain in mice (8, 9), it remains unclear whether similar factors might contribute to weight loss (10) and whether or not these results would translate well to humans (11, 12). Recent feeding studies have shed some light on this issue, demonstrating that humans with higher *Prevotella*-to-*Bacteroides* ratios tend to lose significantly more weight on a high-fiber diet, particularly individuals with low salivary amylase levels (13, 14). Similarly, high cecal *Prevotella* levels have also been shown to improve glucose homeostasis in animal models (15). In humans, it has been shown that a combination of baseline multi-omics features and microbiome data can predict postprandial glycemic responses to various foods (16, 17). Thus, while the exact mechanisms are unknown, the baseline taxonomic composition of the human gut microbiota appears to influence host responses to interventions. However, it is unclear whether or not associations between the gut microbiome and weight loss phenotypes are independent of associations between baseline BMI and the microbiome. For example, individuals with higher baseline BMIs tend to show larger-magnitude drops in BMI upon follow-up, which has been referred to as a "regression-to-the-mean" effect (18). In this study, we set out to understand the possible interactions between baseline BMI, dietary patterns, metabolic health, and gut microbiome profiles and how these factors may be associated with changes in weight and metabolic health following personalized, healthy lifestyle interventions.

## RESULTS

In this study, we leveraged existing data and biobanked samples from the Arivale cohort (see Materials and Methods). Briefly, participants enrolled in a commercial behavioral coaching program run by the former scientific wellness company Arivale, Inc., were paired with a registered dietitian or registered nurse coach. Personalized, telephonic behavioral coaching was provided to each participant on a monthly basis, with email or text communications between coaching calls. This service included longitudinal "deep phenotyping," which involved collecting blood and stool for baseline single nucleotide polymorphism (SNP) genotyping or whole-genome sequencing (blood) and longitudinal clinical labs (blood), metabolomics (blood), proteomics (blood), and 16S amplicon sequencing of the gut microbiome (stool), along with lifestyle questionnaires, body weight measurements, and additional activity-tracking data from wearable devices. Arivale participants undergoing these personalized interventions showed broad improvements across a number of validated health markers, including an average reduction in BMI (19, 20).

We targeted a subset of the ~5,000 Arivale participants to look specifically at weight loss phenotypes during this lifestyle intervention period (Fig. 1A). Briefly, there were 1,252 individuals with blood collected at two time points over the course of a

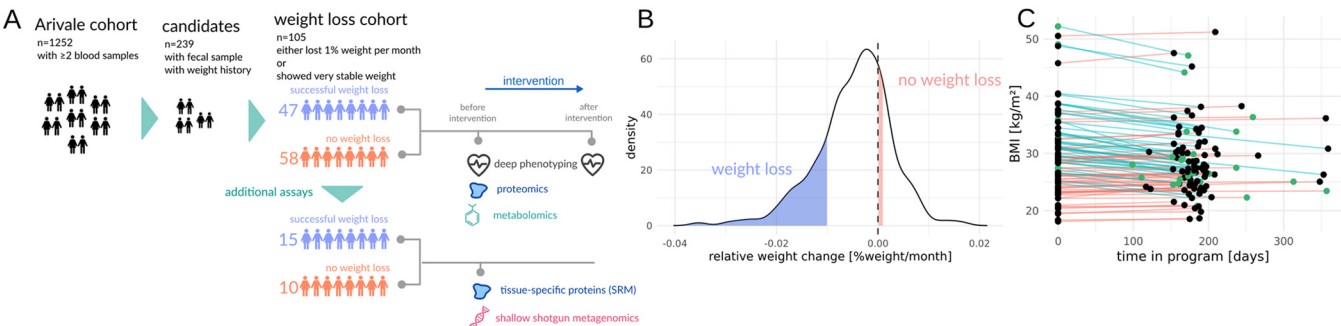

**FIG 1** Study design and cohorts. Schematic showing number of individuals within the Arivale wellness intervention cohort who match our selection criteria for data completeness and "weight loss" and "no weight loss" groups (A). In panels A to C, individuals in the "weight loss" group lost >1% of body weight per month during the program whereas the "no weight loss" group maintained a stable weight, changing less than 0.1%. Distribution of relative weight change for the 239 candidate individuals, with blue area showing individuals who lost >1% of their body weight per month (n = 48) and red area showing individuals who showed no change in weight (n = 57) over the same intervention period (B). Baseline and follow-up BMI values (points from the same individual connected by lines colored by weight loss group: cyan lines denote "weight loss" and red lines denote "no weight loss") for our "weight loss" and "no weight loss" cohorts (n = 105) (C). Green dots in panel C denote individuals with additional proteomic and metagenomic data (n = 25).

year, 239 of whom had a paired stool sample at baseline and longitudinal data on BMI (Fig. 1A to C). We further subdivided these 239 participants by selecting individuals who lost >1% of their body weight per month over a 6- to 12-month period (n = 48) and those who maintained a stable BMI (n = 57) over the same period (Fig. 1B). From this 105-person cohort, another subset of 25 individuals (15 "weight loss" and 10 with "no weight loss") were subselected for additional assays to evaluate whether weight loss responses were associated with (i) concomitant improvements in protein markers of metabolic health and (ii) baseline metagenomic functional gene and taxonomic profiles (Fig. 1A and C). Biobanked fecal samples from this 25-person cohort were used to generate shallow shotgun metagenomes (>2 million reads per sample), in order to obtain gut microbiome functional and taxonomic profiles. Two biobanked plasma samples (taken before and after intervention) taken from each of these 25 individuals were used to generate additional proteomic data on a broad set of obesity and cardiometabolic health markers (see Table S1 in the supplemental material).

In the full cohort of 105 individuals there were no significant differences in age and glucose levels between the "weight loss" and "no weight loss" groups at baseline, but the "weight loss" group had a significantly higher baseline BMI, lower baseline serum high-density lipoprotein (HDL) levels, and slightly lower baseline serum adiponectin levels (Fig. 2A to E). All individuals in the "weight loss" group were considered either overweight or obese (BMI > 25 and 30, respectively), while half of the "no weight loss" group were overweight and the other half were considered normal weight (BMI > 25 and < 25, respectively; Fig. 2A). Across the cohort baseline BMI was significantly

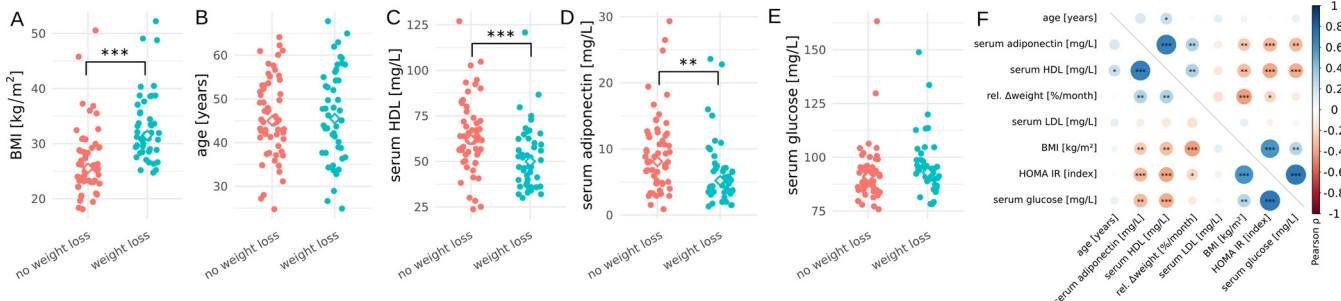

**FIG 2** Baseline metabolic markers in the full cohort (n = 105). Dot plots showing baseline BMI (A), age (B), baseline serum HDL (C), baseline serum adiponectin (D), and baseline serum glucose (E) in the "weight loss" and "no weight loss" groups. In panels A to E, asterisks denote significance under a Welch t test and diamonds denote group medians. Correlation matrix showing Pearson's correlation coefficients between baseline BMI, weight loss, and clinical markers of metabolic health across the entire cohort (F). Asterisks in panel F denote significance under a Pearson moment-product correlation test. In panels A to F, ***, $P < 0.001$; **, $P < 0.01$; *, $P < 0.05$.

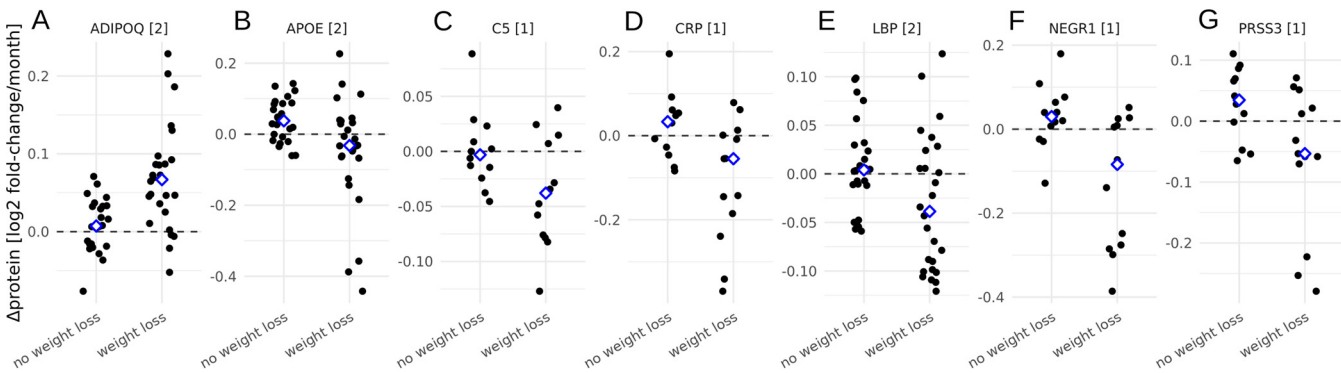

**FIG 3** Additional proteomic markers of metabolic health in the subcohort of 25 individuals. Each plot shows blood proteins that changed significantly in abundance (FDR-corrected *P* < 0.1 when corrected for baseline BMI) between baseline and follow-up sampling in the "weight loss" group, independent of baseline BMI (A to G). Dashed line denotes no change in protein abundance over time, blue diamonds denote the means for the two groups, and numbers in brackets denote how many unique peptide fragments were used to quantify each protein (A to G).

correlated with several baseline metabolic health markers, such as adiponectin, HDL, insulin resistance, and glucose levels (Fig. 2F). We also saw the expected correlation between weight loss and baseline BMI, where those with higher baseline BMIs tended to lose more weight (Fig. 2F), which is often termed the "regression-to-the-mean" effect in weight loss studies (21). Because baseline BMI could potentially mask independent measures associated with weight loss via this regression-to-the-mean effect, we decided to correct all weight loss associations for baseline BMI.

To evaluate whether metabolic health improved in the weight loss group independently of baseline BMI, we used a panel of 22 serum protein markers associated with obesity and metabolic health, measured before and after intervention in our 25-person subcohort. On average, only individuals in the "weight loss" group showed broad improvements in seven blood proteomic markers of metabolic health following the intervention (false-discovery rate [FDR]-corrected analysis of variance [ANOVA] *P* < 0.1, Fig. 3A to G). Specifically, the "weight loss" group showed a marked increase in ADIPOQ (adiponectin) levels, which have previously been negatively associated with BMI and positively associated with fasting (22). The "weight loss" group also showed decreased levels of APOE, C5, CRP, LBP, NEGR1, and PRSS3, which have all been positively associated with obesity, inflammation, and metabolic disorders (Fig. 3B to G) (22–26). Thus, not only did the "weight loss" group reduce their BMI during the intervention period, but they became metabolically and immunologically healthier as well.

We tested for associations between baseline features and weight loss that were independent of baseline BMI, age, and sex (Fig. 4A). Although one might expect baseline phenotypic and dietary factors associated with baseline BMI to have similar associations with changes in BMI (Fig. 4A), we found that these associations were largely independent for blood metabolomics, blood proteomics, 16S genus-level abundances, and dietary patterns in the 105-person cohort (all Pearson's coefficients > 0 or nonsignificant, Fig. 4B to E) and only weakly correlated for metagenome-derived gut microbial species abundances (Pearson rho = −0.22, Fig. 4F) and functional gene abundances (Pearson rho = −0.3, Fig. 4G) in the 25-person subcohort. Thus, phenotypic associations with BMI and weight loss were largely orthogonal (Fig. 4B to G). None of the food frequency measures collected from this cohort were significantly associated with BMI or weight loss (Fig. 4D). Sixty baseline blood- and stool-derived features were independently associated with baseline BMI in the 105-person cohort, including known markers of weight loss and weight gain such as leptin and insulin-like growth factor (Fig. 4B to E) (27, 28). There were no baseline blood metabolites significantly associated with weight change, independent of baseline BMI (Fig. 4B). Only a single protein (KIT ligand) out of 268 baseline proteins tested was independently associated with weight loss resistance (Fig. 4C). The KIT ligand has been reported previously to be associated with obesity and energy expenditure in mice and humans (18, 29, 30). While 6 baseline

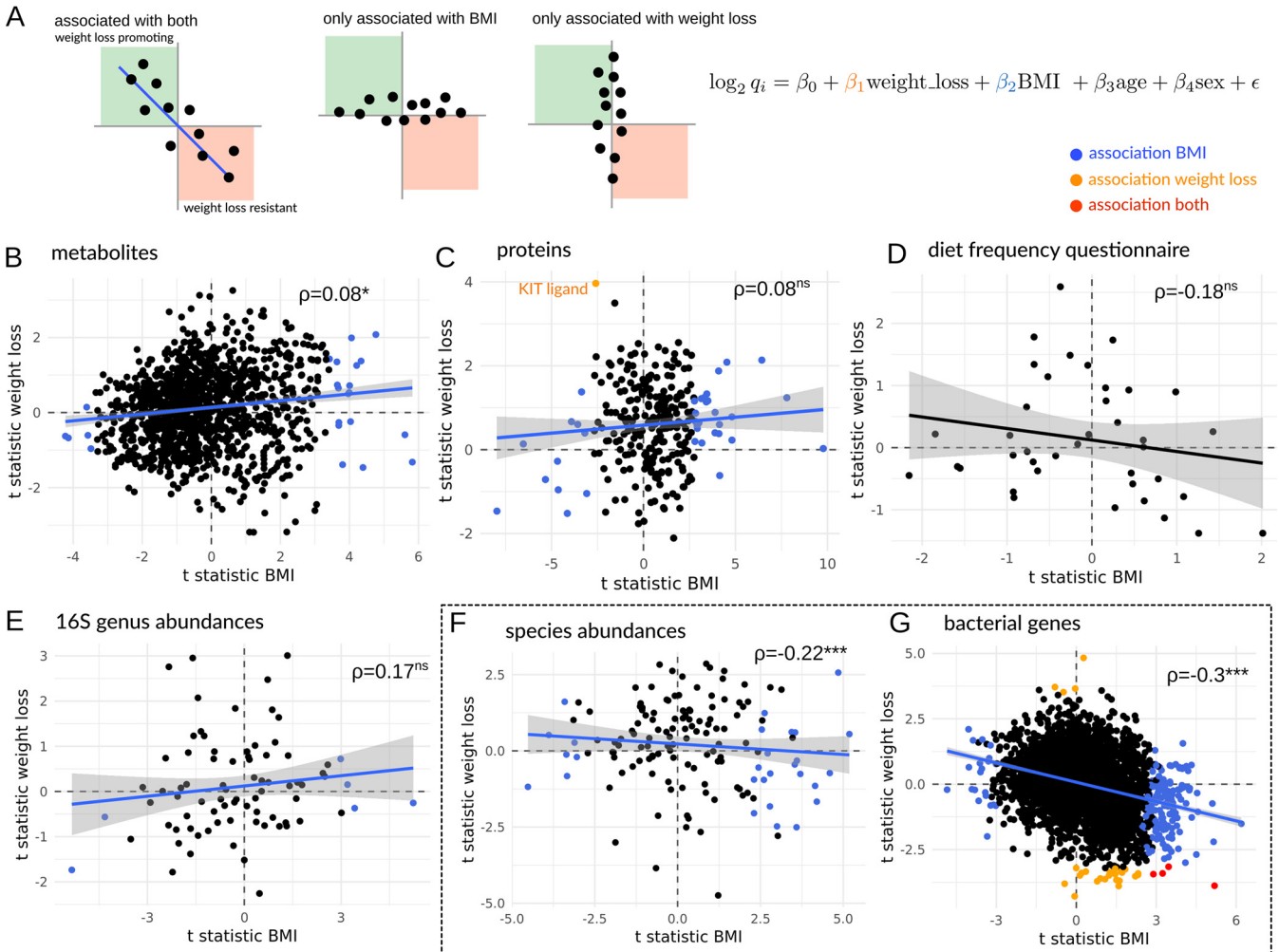

**FIG 4** Associations between baseline multi-omic features, BMI, and weight loss. Biplots show $t$ statistic for features' independent associations with BMI or weight loss, controlling for age and sex (A). Analyses were run separately for baseline blood metabolites (B), baseline blood proteins (C), baseline dietary features (D), baseline 16S gut bacterial genera (E), baseline metagenome-derived gut bacterial species (F), and baseline metagenome-derived gut bacterial functional genes (G). Blue dots denote features significantly associated with BMI only (i.e., independent of weight loss, age, and sex), orange dots denote features significantly associated with weight loss only (independent of BMI, age, and sex), and red dots denote features independently associated with both BMI and weight loss (independent of age and sex). In panels B to G, asterisks denote significance under a Pearson correlation test and ρ denotes the Pearson correlation coefficient between the $t$ statistics for BMI and weight loss (***, $P < 0.001$; **, $P < 0.01$; *, $P < 0.05$; ns, $P > 0.05$). The dashed box around panels F and G denotes metagenomic results from the subcohort of 25 individuals, while the results in panels B to E are from the larger cohort of 105 individuals.

bacterial genera (16S) were associated with baseline BMI in the 105-person cohort, none were independently associated with weight loss (Fig. 4E), and this was consistent with metagenomic species-level results from the 25-person cohort (Fig. 4F). In concordance with a previous study (31), we observed that individuals with higher baseline BMI showed slightly lower metagenomic gene richness, where an increase in 1 BMI unit was associated with a loss of approximately 19 genes ($P = 0.02$, ANOVA corrected for baseline sex and age). Baseline gene richness was not predictive of future weight loss success, when adjusting for baseline BMI (ANOVA $P = 0.93$). However, several of the 2,975 gut bacterial gene clusters included in this analysis showed independent associations with either BMI (177, FDR-corrected $P < 0.05$; see Table S3) or weight loss (27, FDR-corrected $P < 0.05$; see Table S3) in the 25-person cohort, and a few showed independent associations with both baseline BMI and weight loss (4, FDR-corrected $P < 0.05$, Fig. 3G and Table S3). Prior work in a larger cohort of several hundred individuals showed how blood analytes could predict glycemic responders during a clinical weight loss program (5). However, that study did not look at baseline gut microbiomes.

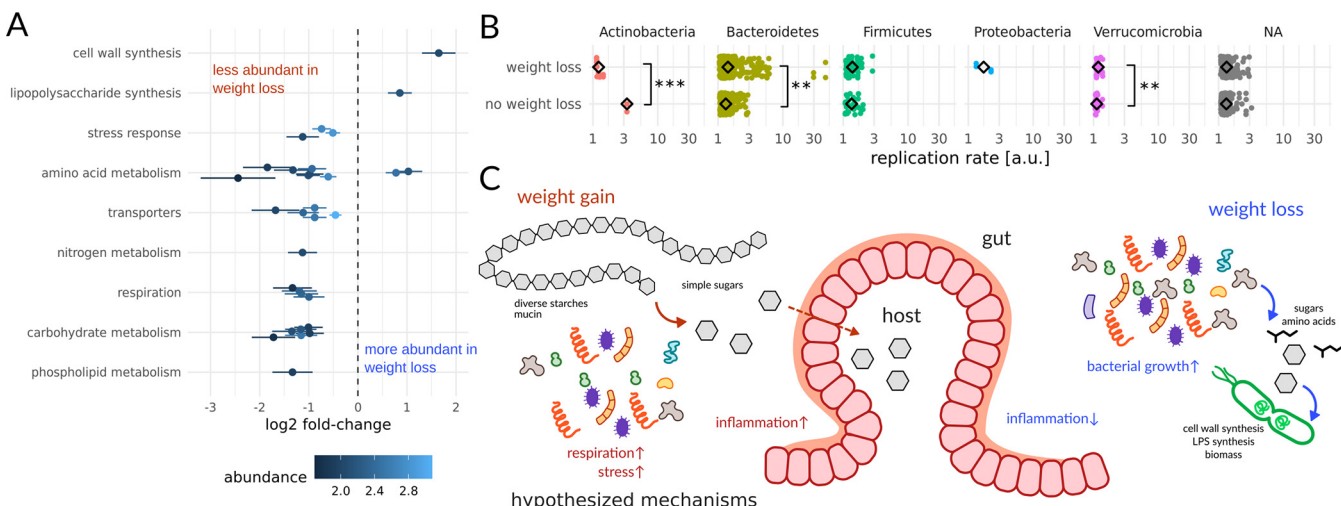

**FIG 5** Metagenomic markers of weight loss in a subcohort of 25 individuals. Metagenomic gene cluster abundances significantly associated with weight loss (independent of baseline BMI, age, and sex), binned into high-level functional categories (A). Average phylum-specific bacterial replication rates estimated from metagenomes show significant differences across weight loss groups (B). In panel B, "NA" denotes contigs without a phylum-level classification (i.e., not enough single-copy phylogenetic marker genes within those contigs to obtain a phylum-level classification) and asterisks denote significance under ANOVAs while correcting for age and baseline BMI (***, $P < 0.001$; **, $P < 0.01$). Schematic of the proposed microbiome-mediated mechanisms involved in weight loss promotion or resistance based on specific metagenomic functions from panel A that were positively or negatively associated with weight loss (C).

Here, we find that the baseline stool metagenomic functional genes show a much larger number of associations with weight loss phenotypes than baseline gut taxonomic, blood proteomic, blood metabolomic, or dietary features (Fig. 4B to G).

In total, 31 baseline gut microbiome functional genes were associated with weight loss, independent of baseline BMI (Fig. 5A). Cell wall and lipopolysaccharide (LPS) synthesis were positively associated with weight loss, which suggested that cell division, biomass production, and Gram-negative bacterial growth potential might be important. To explore this further, we calculated baseline bacterial replication rates directly from metagenome-assembled contigs (32) and found that average replication rates were indeed significantly higher in the "weight loss" group (ANOVA $P = 0.001$, corrected for age and baseline BMI), with Gram-negative *Bacteroidetes* contigs contributing most to this effect (ANOVA $P = 0.002$, Fig. 5B). Most contigs could not be annotated beyond the phylum level, but the fastest-replicating contigs (replication rates > 3) with genus-level annotations belonged to *Prevotella* and were observed only in the weight loss group. Most functional genes were associated with resistance to weight loss: specifically, functions involved in glycan (e.g., glycosyl hydrolases) and protein catabolism, response to stress, peptide antibiotic synthesis, and respiration (Fig. 5A).

## DISCUSSION

Based on our rather preliminary results from this modest-size cohort, we propose a tentative set of hypotheses for how human gut commensals modulate the host's absorption of calories from the diet and potentially impact intestinal inflammation (Fig. 5C). Specifically, we know that the gut microbiota help break down complex, extracellular polysaccharides into simpler sugars that are more readily absorbed by the host. Indeed, we saw that certain CAZy enzyme classes (e.g., GH13, which includes the starch-degrading amylases [see Fig. S3 in the supplemental material]) were enriched in individuals who were resistant to weight loss, independent of baseline BMI. A similar metagenomic increase in bacterial amylase gene frequency has been associated with increased weight gain in mice (33). Furthermore, gut bacterial replication rates were reduced in those who were resistant to weight loss, independent of baseline BMI. Similarly, prior cross-sectional work identified associations between the gut bacterial replication rates from a few taxa and BMI (34). We hypothesize that lower commensal

growth rates may allow the host epithelium to absorb a larger fraction of extracellular polysaccharide breakdown products in the lumen before they can be transformed into less-energy-dense fermentation by-products, like short-chain fatty acids (SCFAs), and bacterial biomass. SCFA production itself can reduce intestinal inflammation (35), which in turn may help to improve metabolic health and better facilitate weight loss (36). Concordantly, we saw reduced levels of circulating inflammation-related proteins in participants who lost weight (Fig. 3A to G). Finally, reduced inflammation could itself promote fermentative metabolism and redox homeostasis in the gut, minimizing oxic stress to strict anaerobes and suppressing respiratory pathways that favor facultative anaerobes (Fig. 3F).

We suggest that dietary energy harvest, host-microbe substrate competition, and modulation of host inflammation by commensal bacteria may be, in part, responsible for determining host responses to weight loss interventions, independent of baseline BMI or metabolic health state. Gut ecosystems optimized for fermentative metabolism and higher bacterial growth rates appear to be conducive to weight loss. Prior work has shown that the higher baseline levels of *Prevotella* can improve weight loss responses to a standardized high-fiber diet (13), and here we found higher baseline *Bacteroidetes* growth rates, driven in part by the genus *Prevotella*, in individuals who lost weight in a commercial wellness program, which often involved suggested increases in dietary fiber and exercise (see Materials and Methods). Recent studies have suggested that one can predict weight loss outcomes following an intervention from baseline 16S rRNA gene community profiles (21, 37). However, these weight loss studies did not correct for baseline BMI, which can act as a significant confounder (e.g., Fig. 2F) due to the regression-to-the-mean effect described above (21, 37). The putative microbiome-centric weight loss mechanisms identified in this study are largely consistent with prior work in nonhuman animal models and in human observational studies, indicating that energy harvest, abundances of glycosyl hydrolase genes, and inflammation are relevant to weight gain and obesity (8, 33, 38, 39).

In summary, our results represent a preliminary set of baseline gut microbiome functional features that are associated with future changes in weight following an intervention, independent of baseline BMI (Fig. 5C). It remains to be seen whether these results will replicate in other cohorts and whether the funneling of dietary starches and fibers into simple sugars accessible to the host, rather than toward conversion into SCFAs, does indeed lead to weight loss resistance. In particular, the current study only looked at baseline dietary patterns and did not track detailed dietary records throughout the full duration of this personalized intervention study. Future studies should capture this longitudinal dietary data in order to better delineate between the influence of dietary variation and baseline gut microbiomes in predicting weight loss responses. Finally, larger interventional trials are needed in humans to further advance our understanding of how our commensal gut microbiota and our lifestyle interact to causally contribute to weight loss. By combining these emerging insights with recently developed models for predicting personalized gut microbiome metabolic outputs (40, 41), we can begin to engineer the functional capacity of our microbiota to optimize the outcomes of dietary and lifestyle interventions.

## MATERIALS AND METHODS

**Arivale cohort and subcohort selection criteria.** Procedures for this study were run under the Western Institutional Review Board (WIRB) with Institutional Review Board (IRB) study number 20170658 at the Institute for Systems Biology and 1178906 at Arivale. The research was performed entirely using deidentified and aggregated data of individuals who had signed a research authorization allowing the use of their anonymized data in research. Per current U.S. regulations for use of deidentified data, informed consent was not required. To be eligible to join the program, participants had to be over 18 years of age, not pregnant, and a resident of any U.S. state except New York. The participants analyzed in this study are the 92% of participants who agreed to research use as of 19 June 2018 and enrolled in the program between July 2015 and March 2018.

Of the ~5,000 Arivale participants who agreed to research use of their data, 1,252 had blood draws at two time points (i.e., a baseline sample and then a follow-up sample at ~6 to 12 months). Of these 1,252 individuals, 239 had follow-up BMI data within the year after the first blood draw and had

biobanked serum and fecal samples available which were sampled within 30 days of each other. We removed individuals with zero variance in weight measurements, which results from digital scales when regular weighing is not performed and the prior weight is reported repeatedly. Relative weight change was calculated as (follow up weight − baseline weight)/months between measurements. The study cohort was then assembled by selecting all individuals who either lost more than 1% of body weight per month ($n = 48$, "weight loss" group) or retained a very stable body weight (gained less than 0.1% of their body weight, $n = 57$, "no weight loss" group) during the lifestyle intervention. A subcohort of 25 of these 105 individuals was selected for additional proteomic and metagenomic assays. Specifically, 15 individuals with the largest declines in weight were used as the "weight loss" group, whereas 10 individuals with the 20 smallest positive weight change values were chosen as the "no weight loss" group (10-person subset was selected to ensure a balanced representation of sexes across groups).

**Arivale behavioral intervention.** Participants who enrolled in the yearlong commercial behavioral coaching program were paired with a registered dietitian or registered nurse coach. Personalized, telephonic behavioral coaching was provided to each participant on a monthly basis, with email or text communications between coaching calls. Each participant's clinical and genetic data were available to them via a web dashboard and mobile app, which they could also use to communicate with their coach and schedule calls or blood draws. Coaches provided specific recommendations to address out-of-range clinical results based on clinical practice guidelines, published scientific evidence, or professional society guidelines. Examples of the evidence behind the coaching recommendations include guidelines from the American Heart Association or American Diabetes Association (42), comprehensive lifestyle interventions such as those developed for the Diabetes Prevention Program (DPP) (43), nutrition recommendations such as those based on the DASH dietary pattern (44) or MIND diet (45), and exercise recommendations from the American College of Sports Medicine (46).

**Blood collection and multi-omic data generation.** Blood draws for all assays were performed by trained phlebotomists at LabCorp or Quest service centers and were scheduled every 6 months, but actual collection times varied. Metabolon conducted their Global Metabolomics high-performance liquid chromatography (HPLC)–mass spectrometry (MS) assays on participant plasma samples. Sample handling, quality control, and data extraction along with biochemical identification, data curation, quantification, and data normalizations have been previously described (47). For analysis, the raw metabolomics data were median scaled within each batch, such that the median value for each metabolite was 1. To adjust for possible batch effects, further normalization across batches was performed by dividing the median-scaled value of each metabolite by the corresponding average value for the same metabolite in quality control samples of the same batch. Missing values for metabolites were imputed to be the minimum observed value for that metabolite. Values for each metabolite were log transformed. Plasma protein levels were measured using three ProSeek proximity extension assay (PEA) panels (cardiovascular II, cardiovascular III, and inflammation arrays) from Olink Biosciences (Uppsala, Sweden), processed, and batch corrected as described previously (47). For analysis, a threshold of less than 5% missing values was set for each protein, which was passed by 263 different analytes. Missing values for the proteins were imputed to be the minimum observed value for that protein.

**Dietary food frequency questionnaires.** Upon sign-up to the Arivale program, individuals filled out extensive questionnaires online. Consumption frequencies for a set of 39 different food entities were presented on an ordinal scale ranging from 0 (no consumption) up to 8 (very frequent consumption). The interpretation of individual consumption levels for each food group can be found in Table S2 in the supplemental material.

**Stool collection and metagenomic data generation.** At-home stool collection kits (DNA Genotek; OMR-200) were shipped directly to participants and then shipped back to DNA Genotek for processing. Microbial DNA was isolated from 200 $\mu$l of homogenized fecal material using the DNeasy PowerSoil Pro extraction kit (Qiagen, Germany) with bead beating in Qiagen Powerbead Pro plates (catalog no. 19311; Qiagen, Germany). Extracted DNA was quantified using the Quant-iT PicoGreen double-stranded DNA (dsDNA) assay kit (Invitrogen, USA), and all samples passed the quality threshold of 1 ng/$\mu$l (range, 8 to 101 ng/$\mu$l).

16S amplicon sequencing was performed as described previously in reference 7. In brief, the 16S V3-V4 region was amplified and sequenced with 300-bp paired-end libraries on an Illumina MiSeq. Samples were demultiplexed using Illumina Basespace (San Diego, CA), yielding the FASTQ files used in this study.

Shallow shotgun sequencing was performed with the BoosterShot service (Corebiome, USA). In brief, single-stranded 100-bp libraries were prepared using an optimized proprietary protocol of the provider (Corebiome, USA) based on the Nextera library prep kit (Illumina, USA) and sequenced on a NovaSeq (Illumina, USA) to a minimum of 2.6 million (2.6M) reads per sample (mean 3.5M, ranging from 2.6M to 4M). Demultiplexing was performed on Basespace (Illumina, USA), yielding the final FASTQ files.

**Anthropometric data.** Height, weight, and waist circumference either were measured at the blood draws (45%) or were self-reported via an online assessment, or through the Fitbit Aria scale. Reference ranges for anthropometric data were defined by U.S. public health guidelines (48).

**SRM of obesity-related proteins.** Serum samples were processed following a previously published protocol that ensured maximum yield of signal (49). We targeted a curated selection of 22 mostly organ-specific proteins with known genetic variants associated with obesity or metabolic syndrome (Table S1). Prepared samples, along with spiked-in heavy-isotope-labeled synthetic standard peptides, were quantified using a triple-quadrupole mass spectrometer (Agilent 6490; Agilent, Santa Clara, CA) with a nano-spray ion source and Chip Cube nano-HPLC. Three to four transitions were monitored for each target peptide (see Table S1). Two micrograms of tryptically digested Mars-14 (Agilent, Santa Clara, CA)

depleted serum was eluted from a high-capacity nano-HPLC chip (160 nl, 150 mm by 75 $\mu$m inside diameter [i.d.]; Agilent, Santa Clara, CA) with a 30-min gradient of 3 to 40% acetonitrile as described previously (49, 50). Raw selective reaction monitoring (SRM) mass spectrometry data were analyzed with the Skyline targeted proteomics environment (51). Each detected peptide was quantified by the light/heavy (L/H) ratio of monitored transitions, after adjusting for the volume of the original serum sample.

**16S amplicon sequencing data processing.** The samples were processed using a customized open-source pipeline previously described in reference 52. Here, individual samples were processed using DADA2 in order to yield individual amplicon sequence variants (ASVs). After merging forward and reverse ASVs, chimeras were removed using the *de novo* algorithm in DADA2, which removed about 17% of all reads as chimeric. Taxonomic names were assigned using the RDP method and using the Silva 16S reference database (version 132). Eighty-nine percent of the total reads could be mapped to at least the genus level this way. The resulting ASV abundance tables, taxonomy assignments, and sample metadata were finally merged into a single phyloseq object that was used for further analysis (53).

**Metagenomics data processing.** Trimming and filtering for the raw sequencing data were performed using FASTP v0.20.1 (54). The first five bases on the 5′ end were trimmed from each read to avoid leftover PCR primers, and each read was furthermore trimmed on the 3′ by the sliding window method with a minimum quality threshold of 20. Abundances of species were obtained using KRAKEN v2.0.9 and BRACKEN v2.6.0 using the default KRAKEN database (55, 56). Contigs were assembled *de novo* with MEGAHIT v1.2.9 with a cross-assembly across all samples. Open reading frames (ORFs) in the resulting contigs were then identified with PRODIGAL v2.6.3 (57). Reads from each sample were then aligned to each contig using MINIMAP2 v2.17, and gene abundances for each sample were quantified with the Expectation-Maximization algorithm from SALMON v3.1.3 (58, 59). The identified ORFs were annotated using the EGGNOG EMAPPER v2.0.1.

Replication rates were inferred using the iRep approach (32). Here, we first aligned the reads for each sample to the full assembled contigs using MINIMAP2 v2.17. Coverage profiles were extracted for all contigs larger than 5,000 bp across bins of a 100-bp width, but only contigs with a minimum length of 11,000 bp and a mean coverage of 2× were used for the iRep inference. Coverage profiles were smoothened using a sliding window mean over 50 bins (5,000-bp window width) before calculating the replication rates using the iRep implementation in mbtools v0.44.14 (https://gibbons-lab.github.io/mbtools). Taxonomic classifications of individual contigs were obtained using CAT v5.1.2 with the default database of single-copy marker genes (60).

**Statistical analyses. (i) SRM data.** Raw SRM abundances from the 25-person cohort were log-transformed, which yielded data that appeared to be normally distributed (as validated by QQ plots). Change in protein abundance across the intervention was then quantified as the difference of protein abundance after intervention and the baseline abundance, yielding log ratios of postintervention versus baseline abundances. Associations with weight loss were obtained by linear regression of the obtained log ratios using the design shown in Fig. 2. Here, assignment to the "weight loss" group was the target covariate, correcting for baseline BMI, age, and sex. Due to the low sample size in the metagenomics cohort, we did not fit interaction terms between sex and weight loss groups as this would have led some coefficients to be estimated from very small cohorts ($n < 6$). False-discovery rates were controlled by adjusting $P$ values using the Benjamini-Hochberg correction.

**(ii) Metabolomic and proteomic data.** For the 105-person cohort, mass spectrometry data from untargeted metabolomics and proteomics data were log-transformed, as this yielded near-normal distributions on QQ plots. Log-abundance values were then used for linear regressions using the design formula shown in Fig. 3A. For each metabolite and protein, we also performed a regression without the "weight loss" group and using the baseline BMI as the target covariate to yield the association strength with BMI. Linear regressions were run using the LIMMA R package without Bayesian shrinkage as this is specific to gene expression data (61). False-discovery rates were controlled by adjusting $P$ values using the Benjamini-Hochberg correction. T-values for each association coefficient were calculated as the ratio of coefficient and estimated coefficient standard deviation obtained from the Fisher matrix of the regression.

**(iii) Diet data.** Responses to food frequency questions were extracted from the 105-person cohort and covered a set of 39 food groups on an ordinal scale ranging from 0 (no consumption) up to 8 (very frequent consumption). The numeric frequency values were used in univariate regression models with the food frequency measure as dependent variable and the same independent variables used in the metabolomic and proteomic data analysis (and shown in Fig. 3A). For each food group we also performed a regression without the "weight loss" group and using the baseline BMI as the target covariate to yield the association strength with BMI. T-values for each association coefficient were again calculated as the ratio of coefficient and estimated coefficient standard deviation obtained from the Fisher matrix of the regression.

**(iv) Metagenomic and 16S data.** Like 16S genus abundances (105 samples), metagenomic species abundances (25 samples) and gene abundances (25 samples) were both obtained from sequencing count data as described above. Each data type was stored as its own phyloseq object. We analyzed both data types (taxon and gene abundances) using negative binomial regressions, which have been shown to fit metagenomic and amplicon sequencing data well (62). This again used the design shown in Fig. 2. However, this time the regressions were performed with negative binomial regression using DESeq2 and using a prior normalization ("poscounts" method in DESeq2) (63). For each microbiome feature (genus, species, or gene) we also performed a regression without the "weight loss" group and using the baseline BMI as the target covariate to yield the association strength with BMI. False-discovery rates were controlled by adjusting $P$ values using the Benjamini-Hochberg correction within each data type.

Pseudo-T-values were calculated as the ratio of coefficient and estimated coefficient standard error obtained from DESeq2. For gene richness estimates all samples were first downsampled to 100,000 total reads with assigned gene clusters. Gene richness was then calculated as the number of observed unique KO term gene clusters in each sample. Regressions were performed with the formulation shown in Fig. 4A and using the gene richness as the response variable.

**Data availability.** Raw metagenomic sequencing data have been deposited on the NCBI Sequence Read Archive (SRA) under BioProject no. PRJNA748449. SRM data can be found on the GitHub repository associated with this study (https://github.com/gibbons-lab/weight_loss_2019). The Institute for Systems Biology manages all Arivale data requests for nonprofit research purposes and will grant access to qualified researchers. Data requests should be sent to A.T.M. (andrew.magis@isbscience.org). The full workflow used to process the metagenomic data is provided as a Nextflow pipeline at https://github.com/Gibbons-Lab/pipelines/tree/master/shallow_shotgun. All analyses can be found in Rmarkdown notebooks, which allow the reproduction of all analyses and figures in this paper (https://github.com/gibbons-lab/weight_loss_2019). Specialized functions, such as the specific implementation for calculating replication rates or association analyses, can be found in a dedicated R package along with documentation at https://github.com/gibbons-lab/mbtools.

## SUPPLEMENTAL MATERIAL

Supplemental material is available online only.

**FIG S1**, PDF file, 1 MB.
**FIG S2**, PDF file, 0.8 MB.
**FIG S3**, PDF file, 0.05 MB.
**TABLE S1**, CSV file, 0 MB.
**TABLE S2**, CSV file, 0.01 MB.
**TABLE S3**, CSV file, 0.2 MB.

## ACKNOWLEDGMENTS

This work was supported by an Institute for Systems Biology Innovator Award (principal investigators [PIs] C.D. and S.Q.). S.M.G. and C.D. were supported by the Washington Research Foundation Distinguished Investigator Award and startup funds from the Institute for Systems Biology. The funders had no role in designing, carrying out, or interpreting the work presented in the manuscript.

Contributions: C.D., S.Q., S.M.G., and L.H. conceptualized the study; C.D. and S.P. acquired and managed samples for metagenomics; S.Q., Y.Z., and L.T. performed the SRM experiments; J.C.L., A.T.M., and N.D.P. ran and supervised the healthy lifestyle intervention and obtained the deep phenotyping data from the Arivale cohort; C.D., S.P., and S.M.G. analyzed the data; S.M.G. and C.D. wrote the original draft. All authors reviewed and edited the manuscript.

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
