## [Reviewer comments · mSystems]

Baseline gut metagenomic functional gene signature associated with variable weight loss responses following a healthy lifestyle intervention in humans

Christian Diener, Shizhen Qin, Yong Zhou, Sushmita Patwardhan, Li Tang, Jennifer Lovejoy, Andrew Magis, Nathan Price, Lee Hood, and Sean Gibbons

Corresponding Author(s): Sean Gibbons, Institute for Systems Biology

Review Timeline:

Submission Date:	July 23, 2021
Editorial Decision:	August 14, 2021
Revision Received:	August 18, 2021
Accepted:	August 23, 2021

Editor: Danilo Ercolini

Reviewer(s): Disclosure of reviewer identity is with reference to reviewer comments included in decision letter(s). The following individuals involved in review of your submission have agreed to reveal their identity: Henrik Munch Roager (Reviewer #1)

Transaction Report:

DOI: <https://doi.org/10.1128/mSystems.00964-21>

August 14, 2021

Prof. Sean M. Gibbons
Institute for Systems Biology
401 Terry Ave. N
Room 359
Seattle, WA 98109

Re: mSystems00964-21 (Baseline gut metagenomic functional gene signature associated with variable weight loss responses following a healthy lifestyle intervention in humans)

Dear Prof. Sean M. Gibbons, dear Sean,

Thank you for submitting your manuscript to mSystems. We have completed our review and I am pleased to inform you that, in principle, we expect to accept it for publication in mSystems. However, acceptance will not be final until you have adequately addressed the reviewer comments.

Below you will find instructions from the mSystems editorial office and comments generated during the review.

Preparing Revision Guidelines

For complete guidelines on revision requirements for your article type, please see the journal Article Types requirement at <https://journals.asm.org/journal/mSystems/article-types>. **Submissions of a paper that does not conform to mSystems guidelines will delay acceptance of your manuscript.**

Sincerely,

Danilo Ercolini

Editor, mSystems

Journals Department
Reviewer comments:

Reviewer #1 (Comments for the Author):

I think the authors have significantly improved the manuscript and have addressed my main concerns. I think it is now presented in a more clear and balanced way. The authors increased the sample size for many of the results presented. However, the key results on the differences in 31 microbiome functional features (out of 2975) between 'weightloss group' and controls still rely on a subset of only 25 individuals. Therefore, I am also pleased to see that the language has been adjusted throughout the manuscript to better reflect the preliminary nature of the results.

Furthermore, I am pleased to see that the authors now discuss the baseline dietary data. However, the lack of proper dietary information (including assessment of changes in dietary habits) is a major limitation of the study given its emphasis on weight loss. I would suggest that the authors acknowledge this in the discussion and also states that it cannot be ruled out that the weight loss and differences in the gut microbiome functional features between weight loss and control group were mediated by changes in diet, since only the baseline diet was recorded.

The authors write in the 'importance section' (line 44-46) that their results provide mechanistic insights. I think the authors should temper/re-phrase this, as the results are purely based on associations.

Reviewer #2 (Comments for the Author):

The revised manuscript adequately addresses most of my concerns:

A. The introduction provides a helpful review of the state of the field and indicates two knowledge gaps: what are the mechanisms through which the microbiome might contribute to efficacy of weight loss interventions, and whether the microbiome's effect is independent from that of baseline BMI. This is helpful for understanding the aims of this work in the context of previous literature. The

edits in the results section and figures are also helpful. I think the revised discussion really helps placing the authors' findings in the context of previous works.

B. The authors have performed additional analyses on the larger sub-cohort of n=105 individuals; This helps to abate one of my concerns when reading the original submission, which was related to the cohort size and whether the study is sufficiently powered to detect significant effects.

C. Additional health related parameters other than BMI were added or are better presented.

My only remaining concern relates to the potential contribution of the participants' diet during the study (and at baseline) to the observed effects, that are only partly addressed in the revised manuscript. The FFQ data included in the revised manuscript is not granular enough to rule out an effect of habitual diet on the observed effects, and there is no dietary data during the intervention. Since I do find this manuscript of interest, and if I understand correctly additional dietary data are not available, and since the current analysis, albeit its limitations, did not find contribution of the baseline diet to the observed effect - I would only ask that the authors add this limitation to the text / discussion, i.e. that due to the limited dietary information and no information on diet during the intervention, they cannot completely rule out that some of the effects on the metabolic parameters / microbiome / other omics are explained by diet.

Additional minor comments

1. In the legend for figure 1, it would be helpful to add how "no change" in weight was defined (currently only mentioned in the methods).
2. Previous studies have suggested that species or gene richness are associated with weight loss - it would be interesting to explore this association with the current cohort.
3. In figure 1C, it would be helpful to color the relevant lines in blue (weight loss) and red (no weight loss) rather than in green).
4. I'm missing here analysis of the results considering sex as a biological variable.
5. It would be helpful to provide the list of the 31 baseline gut microbiome functional genes that are associated with weight loss (is that supposed to be table 3? It was not included in the submission, but is mentioned in the text).

The revised manuscript adequately addresses most of my concerns:

- A. The introduction provides a helpful review of the state of the field and indicates two knowledge gaps: what are the mechanisms through which the microbiome might contribute to efficacy of weight loss interventions, and whether the microbiome's effect is independent from that of baseline BMI. This is helpful for understanding the aims of this work in the context of previous literature. The edits in the results section and figures are also helpful. I think the revised discussion really helps placing the authors' findings in the context of previous works.
- B. The authors have performed additional analyses on the larger sub-cohort of n=105 individuals; This helps to abate one of my concerns when reading the original submission, which was related to the cohort size and whether the study is sufficiently powered to detect significant effects.
- C. Additional health related parameters other than BMI were added or are better presented.

My only remaining concern relates to the potential contribution of the participants' diet during the study (and at baseline) to the observed effects, that are only partly addressed in the revised manuscript. The FFQ data included in the revised manuscript is not granular enough to rule out an effect of habitual diet on the observed effects, and there is no dietary data during the intervention. Since I do find this manuscript of interest, and if I understand correctly additional dietary data are not available, and since the current analysis, albeit its limitations, did not find contribution of the baseline diet to the observed effect – I would only ask that the authors add this limitation to the text / discussion, i.e. that due to the limited dietary information and no information on diet during the intervention, they cannot completely rule out that some of the effects on the metabolic parameters / microbiome / other omics are explained by diet.

Additional minor comments

1. In the legend for figure 1, it would be helpful to add how "no change" in weight was defined (currently only mentioned in the methods).
2. Previous studies have suggested that species or gene richness are associated with weight loss – it would be interesting to explore this association with the current cohort.
3. In figure 1C, it would be helpful to color the relevant lines in blue (weight loss) and red (no weight loss) rather than in green).
4. I'm missing here analysis of the results considering sex as a biological variable.
5. It would be helpful to provide the list of the 31 baseline gut microbiome functional genes that are associated with weight loss (is that supposed to be table 3? It was not included in the submission, but is mentioned in the text).

August 20, 2021

Prof. Sean M. Gibbons
Institute for Systems Biology
401 Terry Ave. N
Room 359
Seattle, WA 98109

Re: mSystems00964-21R1 (Baseline gut metagenomic functional gene signature associated with variable weight loss responses following a healthy lifestyle intervention in humans)

Dear Prof. Sean M. Gibbons:

Your manuscript has been accepted, and I am forwarding it to the ASM Journals Department for publication. For your reference, ASM Journals' address is given below. Before it can be scheduled for publication, your manuscript will be checked by the mSystems senior production editor, Ellie Ghatineh, to make sure that all elements meet the technical requirements for publication. She will contact you if anything needs to be revised before copyediting and production can begin. Otherwise, you will be notified when your proofs are ready to be viewed.

As an open-access publication, mSystems receives no financial support from paid subscriptions and depends on authors' prompt payment of publication fees as soon as their articles are accepted. =

Publication Fees:

We recognize that the video files can become quite large, and so to avoid quality loss ASM suggests sending the video file via <https://www.wetransfer.com/>. When you have a final version of the video and the still ready to share, please send it to Ellie Ghatineh at eghatineh@asmusa.org.

Sincerely,

Danilo Ercolini
Editor, mSystems

Journals Department
Table S2: Accept
Figure S3: Accept
Figure S1: Accept
Table S1: Accept
Figure S2: Accept
Table S3: Accept